# MiR-150-5p Overexpression in Triple-Negative Breast Cancer Contributes to the In Vitro Aggressiveness of This Breast Cancer Subtype

**DOI:** 10.3390/cancers14092156

**Published:** 2022-04-26

**Authors:** Bruna M. Sugita, Yara Rodriguez, Aline S. Fonseca, Emanuelle Nunes Souza, Bhaskar Kallakury, Iglenir J. Cavalli, Enilze M. S. F. Ribeiro, Ritu Aneja, Luciane R. Cavalli

**Affiliations:** 1Research Institute Pele Pequeno Príncipe, Faculdades Pequeno Príncipe Curitiba, Curitiba 80250-060, Brazil; bruna.sugita@aluno.fpp.edu.br (B.M.S.); aline.fonseca@pelepequenoprincipe.org.br (A.S.F.); emanuelle.souza@aluno.fpp.edu.br (E.N.S.); 2Department of Oncology, Lombardi Comprehensive Cancer Center, Georgetown University Medical Center, Washington, DC 20007, USA; yararodriguez2022@u.northwestern.edu; 3Genetics Post-Graduation Program, Department of Genetics, Federal University of Paraná, Curitiba 81530-000, Brazil; cavalli@ufpr.br (I.J.C.); eribeiro@ufpr.br (E.M.S.F.R.); 4Department of Pathology, Georgetown University Medical Center, Washington, DC 20007, USA; kallakub@georgetown.edu; 5Department of Clinical and Diagnostic Sciences, School of Health Professions, University of Alabama at Birmingham, Birmingham, AL 35294, USA; raneja@uab.edu

**Keywords:** miR150-5p, triple-negative breast cancer, TNBC, aggressive tumor phenotypes

## Abstract

**Simple Summary:**

Triple-negative breast cancer (TNBC) is a clinically aggressive type of breast cancer. MicroRNAs (miRNAs) are small molecules that regulate the expression of genes involved in tumor cell signaling. The miR-150-5p is frequently deregulated in cancer, with expression and mode of action varying according to the cancer type. In this study, we investigated the expression levels of miR-150-5p in TNBC, its association with clinical and pathological features of patients, and its role in modulating TNBC cell proliferation, migration, and drug resistance. Our results suggest that miR-150-5p is highly expressed in TNBC and that miR-150-5p expression levels are associated with tumor grade, patient survival, and ethnicity. Our findings also indicate that miR-150-5p contributes to the aggressive phenotypes of TNBC cells in vitro.

**Abstract:**

MiR-150-5p is frequently deregulated in cancer, with expression and mode of action varying according to the tumor type. Here, we investigated the expression levels and role of miR-150-5p in the aggressive breast cancer subtype triple-negative breast cancer (TNBC). MiR-150-5p expression levels were analyzed in tissue samples from 113 patients with invasive breast cancer (56 TNBC and 57 non-TNBC) and 41 adjacent non-tumor tissues (ANT). Overexpression of miR-150-5p was observed in tumor tissues compared with ANT tissues and in TNBC compared with non-TNBC tissues. MiR-150-5p expression levels were significantly associated with high tumor grades and the Caucasian ethnicity. Interestingly, high miR-150-5p levels were associated with prolonged overall survival. Manipulation of miR-150-5p expression in TNBC cells modulated cell proliferation, clonogenicity, migration, and drug resistance. Manipulation of miR-150-5p expression also resulted in altered expression of its mRNA targets, including epithelial-to-mesenchymal transition markers, *MYB*, and members of the SRC pathway. These findings suggest that miR-150-5p is overexpressed in TNBC and contributes to the aggressiveness of TNBC cells in vitro.

## 1. Introduction

Triple-negative breast cancer (TNBC) is a clinically aggressive breast cancer subtype that contributes to the patient’s unfavorable prognosis, resistance to therapy, and a high risk to develop metastatic lesions [1]. TNBCs exhibit the classical “hallmarks” of invasive tumors, including high levels of genomic instability, impaired function of BRCA1 and estrogen signaling, and high frequency of mitotic and angiogenic events [2,3,4].

MicroRNAs (miRNAs) are a class of endogenous non-coding RNA molecules that regulate key biological processes, including cell development and malignant transformation. In cancer, miRNAs can act as oncogenes or tumor suppressor genes by regulating the expression of genes that control cell proliferation, differentiation, apoptosis, invasion, and metastasis [5,6,7]. Therefore, miRNAs have emerged as novel and promising therapeutic targets in cancer [8,9].

MiR-150-5p, a miRNA localized on chromosome 19q13, regulates the gene expression of several cancer driver genes. Deregulation of miR-150-5p expression has been observed in different cancers, with patterns that vary according to the tumor type. MiR-150-5p is downregulated in hematologic diseases and in head and neck, liver, cervical, colorectal, and ovarian carcinomas, suggesting a tumor suppression mode of action in these cancers [10,11,12,13,14,15,16,17,18,19]. In contrast, miR-150-5p is upregulated in lung and gastric cancers, promoting tumorigenesis [20,21,22,23,24]. Results on the expression and role of miR-150-5p in breast cancer (including TNBC) are conflicting; both upregulation and downregulation of miRNA-150-5p have been reported in breast tumor tissues compared to normal or adjacent non-tumor (ANT) tissues [25,26,27,28,29].

In vitro studies on TNBC cell lines have demonstrated that miR-150-5p overexpression is present in cells with more aggressive characteristics, such as the ones with high proliferation rates, the ability to form colonies, migrate, invade, and develop distant metastasis [28,29,30]. However, other studies have shown that miR-150-5p overexpression inhibits breast tumorigenesis [29,31]. Distinct mechanisms have been proposed to underlie the varied roles of miR-150-5p in modulating these phenotypes. These mechanisms include the direct or indirect regulation of its mRNA targets, such as *MYB*, *HGMA2*, *TP53*, *SRC1N1*, and *ZEB1* [25,26,28], as well as its interaction with long non-coding RNAs [25,26,31,32].

In this study, we investigated the expression levels of miR-150-5p in clinically well-annotated TNBC specimens and its role in conferring aggressive tumor phenotypes in TNBC cells. 

## 2. Materials and Methods

### 2.1. Clinical Samples

Non-identified breast cancer patients’ samples of formalin-fixed paraffin-embedded (FFPE) material was obtained from 113 invasive breast carcinoma tissues and 41 adjacent non-tumor (ANT) tissues. The samples were retrospectively collected from the Histopathology Shared Resources (HSTR) at the Lombardi Comprehensive Cancer Center, Georgetown University, under IRB approval (IRB#1992-048). The tumors were classified as TNBC (*n* = 56) or non-TNBC (*n* = 57) subtype by the expression of ER, PR, and HER2 protein markers according to international guidelines [33,34]. Paired ANT tissues were obtained from TNBC (*n* = 21) and non-TNBC (*n* = 20) FFPE sections after pathological evaluation.

The patients’ clinical data and tumors’ histopathological parameters were retrieved by the HSTR personnel. This information included age at diagnosis, ethnicity (self-reported), size (cm), histology, and tumor grade, and lymph node and distant metastasis involvement.

The mean age at diagnosis was 51.7 ± 11.64 (range, 30–70 years) in the TNBC group and 53.3 ± 12.24 (range, 32–88 years) in the non-TNBC group. Tumors sized ≤2 cm were present in 58.7% of patients in the TNBC group and 64.7% in the non-TNBC group; tumors sized between 2 cm and 5 cm were present in 39.1% and 31.4% of patients, respectively. Tumors >5 cm were observed in only one patient in the TNBC group and two patients in the non-TNBC group. All patients in the TNBC group had invasive ductal carcinomas and tumor grade 2/3. In contrast, patients in the non-TNBC group presented both ductal (96%) and lobular invasive carcinomas (12.3%), and only 39.6% had grade 2/3 tumors. Lymph node metastasis was observed in 40.5% of patients in the TNBC group and 50% of those in the non-TNBC group; distant metastases were observed in 6.9% and 18.2% of patients, respectively. Most patients in both groups were African Americans. Clinical outcomes (alive/deceased) were known only for 24 patients with TNBC. The median follow-up time was 128.4 ± 57.98 months (range, 23.9–252.8 months); 20 patients were alive, and four were deceased. Tumor histology and grade differed significantly between patients with TNBC and those with non-TNBC (Table 1).

### 2.2. Breast Cell Lines

One ER+ breast cancer cell line (MCF-7), six TNBC cell lines (BT549, HCC1806, MDA-MB-157, MDA-MB-231, MDA-MB-453, and MDA-MB-468), and one non-tumorigenic human breast epithelial cell line (MCF-10A) were acquired from the Tissue Culture Shared Resource (TCSR) at the Lombardi Comprehensive Cancer Center, Georgetown. The cell lines were authenticated by STR allelotype analysis, following the International Cell Line Authentication Committee (ICLAC) guidelines [35] (Title: Guide to Human Cell Line Authentication. Available online: https://iclac.org/resources/human-cell-line-authentication/, accessed on 1 June 2018). Cells were cultivated in Roswell Park Memorial Institute Medium (RPMI) 1640 or Dulbecco’s Modified Eagle’s Medium (DMEM) media supplemented with 10% Fetal Bovine Serum (FBS), 5% Penicillin/Streptomycin (P/S), and 1% Fungizone in T75 culture flasks at 37 °C and in 5% CO_2_ conditions. Cells were harvested for the functional assays in a confluence of 70% to 80%. Considering the baseline levels of the HCC1806 and MDA-MB-231 cells, their transfection efficiency using our reverse transfection protocol for both miR-150-5p mimic and inhibitor, and their high metastatic potential, these cells were selected for the functional assays performed. 

### 2.3. RNA Isolation

Areas of tumor and adjacent normal tissue (ANT) were microdissected from 10-μm tissue sections of FFPE tumor blocks, after pathology inspection, according to Torresan et al. [36]. The RecoverAll Total Nucleic Acid Isolation kit (Thermo Fisher Scientific, Waltham, MA, USA), was used to isolate RNA and the 2100 Bioanalyzer (Agilent Technologies Inc., Santa Clara, CA, USA), and a NanoDrop spectrophotometer (Thermo Fisher Scientific, Waltham, MA, USA) were used to assess RNA quality and quantity, respectively. RNA was isolated from breast cell lines using the same procedure.

### 2.4. Real-Time Quantitative PCR (RT-qPCR)

The expression levels of miR-150-5p were assessed in clinical samples and in the cell lines by RT-qPCR using a Taqman MicroRNA Assay for hsa-miR-150-5p (Assay#000473, Life Technologies, Carlsbad, CA, USA), following the manufacturer’s protocol. RNU48 was used as endogenous control. MiRNA expression levels were calculated using the 2^−ΔΔCt^ method [37].

### 2.5. Reverse Transfection Assays

The HCC1806 and MDA-MB-231 cells were subjected to the manipulation of miR-150-5p expression levels using mirVanaTM miRNA mimic (miR-150-5p-mimic) and inhibitor (miR-150-5p-inh) assays (Life Technologies, Carlsbad, CA, USA). In all the transfection assays a negative control (NC) was included. The viability of the cells to different concentrations of lipofectamine was assessed prior to the transfection assays. Reverse transfection was performed in triplicate, according to our previous protocol [38]; inhibition and overexpression of miR-150-5p were achieved using a final concentration of 50 nM after 48 h of transfection. The efficiency of transfection was determined by the expression levels of miR-150-5p by RT-qPCR in the TNBC transfected cells in relation to the NC.

### 2.6. Cell Proliferation Assay

Cell proliferation was assessed in the TNBC transfected cells using the Cell Titer Aqueous Solution Cell Proliferation Assay (MTS; Promega, Madison, WI, USA). HCC1806 and MDA-MB-231 transfected cells (4 × 10^3^) were plated in 96-well culture plates (20 mL/well) and exposed to the Cell Titer Solution. Proliferation rate was assessed at 24, 48, 72, and 96 h using an ELISA reader by measuring optical absorbance at 490 nm. The period of 48 h after transfection was considered Day 0. Experiments were performed in triplicate.

### 2.7. Clonogenic Assay

A clonogenic assay was performed to assess colony formation in the TNBC transfected cells according to Franken et al. (2006) [39] with modifications. HCC1806 cells (1 × 10^3^ cells) were trypsinized 48 h after transfection, seeded in 6-well plates and placed at 37 °C for colony formation. The medium was replaced at seven days and after 14 days the cells were fixed with methanol and acetic acid (3:1 ratio) and stained with 0.5% crystal violet (Sigma Aldrich, St. Louis, MO, USA). The colonies (considered of clusters with >50 cells) of triplicate experiments were counted under a light microscope.

### 2.8. Wound Healing Assay

HCC1806 and MDA-MB-231 cells (3 × 10^4^ cells/well) were grown in medium with low concentration of fetal bovine serum and in the presence of mitomycin C (MMC) a mitotic inhibitor, as recommended to suppress proliferation in the wound healing assays [40,41,42,43]. The cells were then seeded at 3 × 10^4^ cells/well and reverse transfected with miR-150-5p mimic, miR-150-5p inhibitor, and NC directly into the 2-well culture-inserts in µ-Dish 35 mm (Ibidi GmbH, Gräfelfing, Germany), according to our previous protocol [38]. After 48 h of transfection, the inserts were removed, and the cells were washed with Phosphate Buffered Saline (PBS). Migration activity was evaluated by the gap distance between the inserts, measured in the images captured at 0 h and after 24 h and analyzed using ImageJ software [44].

### 2.9. Cytotoxic Assays

HCC1806 transfected cells were plated (3 × 10^4^ cells/well) and exposed to 500 nM of doxorubicin. The MTT assay was used per standard protocols. Cell viability was assessed at 24, 48, 72, and 96 h after treatment, using an ELISA reader by measuring optical absorbance at 490 nm. Experiments were performed in triplicate.

### 2.10. Western Blotting

Protein lysates of HCC1806 and MDA-MB-231 transfected cells were obtained according to our previous protocol [38]. Briefly, the cells were lysed in Radio-Immunopreciptation Assay (RIPA) buffer (Invitrogen, Waltham, MA, USA) with protease inhibitor cocktail (Invitrogen, MA, USA). The proteins were subjected to Bolt-PAGE Novex 10% Bis-Tris gel (Thermo Fisher Scientific, Eugene, OR, USA), using the iBlot 2 Gel Transfer Device (Thermo Fisher Scientific, Eugene, OR, USA). The primary antibodies used were Epithelial-Mesenchymal Transition (EMT) (Antibody Sampler Kit, Cell Signaling Technology, Inc., Danvers, MA, USA): SLUG (C19G7; dilution 1:1000), SNAIL (C15D3; dilution 1:1000), VIM (D21H3; dilution 1:1000), and ZEB 1 (D80D3; dilution 1:2000); c-MYB (PA5-101014; dilution1:300; Invitrogen-Thermo Fisher Scientific, Waltham, MA, USA); p-Erk (1/2) (L34F12; dilution 1:1000; Cell Signaling Technology (CST), MA, USA), p-Src (Y416) (dilution1:1000; CST); p-Scr (Y517) (D7F2Q; dilution 1:1000; CST); GAPDH (D46CR; dilution 1:1000, CST). AP-conjugated secondary antibody (dilution 1:10,000) was obtained from Life Technologies. The proteins were visualized using SuperSignal West Pico Enhanced Chemiluminescent (ECL) (Thermo Fisher Scientific, Waltham, MA, USA) and Amersham Imager 600 (Amersham Biosciences, Amersham, Buckinghamshire, UK). Images were analyzed using ImageJ software [44].

### 2.11. The Cancer Genome Atlas Data Processing and Analysis

MiR-150-5p expression was assessed in The Cancer Genome Atlas (TCGA)-Breast Carcinoma (BRCA) dataset (2016_01_28 data version) obtained using the GDAC FireBrowse TCGA portal [45] (“Broad GDAC Firehouse. Available online: http://firebrowse.org/ accessed on 17 February 2022”). Log2-normalized mature miR-150-5p counts were obtained for 755 primary tumors (PT) and 87 normal tissues (NT). Two hundred thirty-eight PT samples had information on ER, PR, and HER2 expression and were classified as TNBC (*n* = 46) or non-TNBC (*n* = 192). The Student’s *t*-test was used to compare miR-150-5p expression levels, using *p* < 0.05 as a threshold of significance. Association with survival was obtained using the aggregated breast cancer clinical studies extracted from TCGA and Molecular Taxonomy of Breast Cancer International Consortium (METABRIC) databases (selected for the TNBC subtype) available at the KM Plotter [46] (“Kaplan Meier Plotter. Available online: http://kmplot.com/analysis/ accessed on 1 February 2022”). The data was displayed considering hazard ratios (HRs), confidence intervals, and log-rank *p* values.

### 2.12. Network of miR-150-5p and Functionally Validated Target mRNAs

The online tool miRTargetLink 2.0 [47] (Saarland University, Saarbrucken, Germany (“miRTargetLink 2.0. Available online: https://ccb-compute.cs.uni-saarland.de/mirtargetlink2 accessed on 16 February 2022”) was used to obtain a list of functionally validated target genes of miR-150-5p. Only strong interactions were considered in this analysis. The STRING tool [48] (“STRING. Available online: https://string-db.org accessed on 16 February 2022”) was used to determine protein-protein interactions between the target genes, and a network was constructed on Cytoscape 3.9.1 (Cytoscape Team, Seattle, WA, USA) [49] using the stringApp.

### 2.13. Statistical Analysis

MiRNA and mRNA expression data were obtained using SDS 2.4 software (Applied Biosystems, Waltham, MA, USA), and the relative expression analysis was performed by the 2^−∆∆Ct^ method [37]. Data were presented as the mean and standard error of the mean (SEM) from three independent experiments. A Fisher’s exact test was used to compare age at diagnosis (≤53, >53), tumor size (≤2 cm, >2 and <5 cm, >5 cm), tumor histology (ductal, lobular), tumor grade (1/2, 3/4), ethnicity (Caucasian, African American), and lymph node and distant metastasis status (positive, negative) between groups. MiR-150-5p expression levels between TNBC, non-TNBC, and ANT groups were compared using the Wilcoxon signed-rank test. The relationship between miR-150-5p levels and clinical and histological parameters was assessed using the Mann–Whitney *U* test. Survival analysis was performed using Kaplan–Meier and log-rank tests. Statistical analyses were carried out using GraphPad Prism version 8 (San Diego, CA, USA), using *p* values < 0.05 as statistically significant.

## 3. Results

### 3.1. MiR-150-5p Is Significantly Upregulated in Tumor Breast Tissues and in TNBC

The expression levels of miR-150-5p in breast tumor tissues (*n* = 113) and ANT tissues (*n* = 41) were assessed by RT-qPCR. MiR-150-5p was significantly upregulated in tumor tissues compared to ANT tissues (logFC = 3.55, *p* < 0.0001; Figure 1A). Consistently, analysis of miR-150-5p expression in paired samples showed that miR-150-5p was upregulated in TNBC and non-TNBC tumor tissues compared with their respective ANT tissues (ANT vs. non-TNBC: logFC = 1.97, *p* < 0.0001, *n* = 20 pairs; ANT vs. TNBC: logFC = 2.25, *p* < 0.0001, *n* = 21 pairs; Figure 1B,C). TNBCs (*n* = 56) presented higher expression levels of miR-150-5p compared to non-TNBCs (*n* = 57) (logFC = 0.93, *p* < 0.01; Figure 1D). MiR-150-5p expression levels could discriminate tumor tissues from ANT tissues with an area under the curve (AUC) value of 0.8355 (95% confidence interval [CI], 0.7744–0.8967). Additionally, miR-150-5p levels could discriminate between paired TNBC and ANT tissues (AUC = 0.7982; CI 95%, 0.6494–0.9489), as well as between non-TNBC and TNBC tissues (AUC = 0.7750; CI 95%, 0.6203–0.9297; Figure 1A–D).

These results are in agreement with the miR-seq data from the TCGA-BRCA dataset, which showed significantly higher miR-150-5p expression levels in tumor tissues (*n* = 775) than in non-tumor tissues (*n* = 87) (*p* = 0.0390) and in TNBC tissues (*n* = 46) than in non-TNBC tissues (*p* = 0.0015) (Figure 1E).

### 3.2. MiR-150-5p Expression Levels Are Associated with Tumor Grade, Ethnicity, and Overall Survival in TNBC

The levels of miR-150-5p were significantly associated with tumor grade in the entire cohort (*p* = 0.0039) and in the TNBC subgroup (*p* = 0.0453), with higher expression levels in grade 3/4 tumors (Figure 2A). A significant association was also observed with the patients’ ethnicity in the entire cohort (*p* < 0.0001), as well as in TNBC (*p* = 0.0107) and non-TNBC subgroups (*p* < 0.0001), with Caucasians presenting higher miR-150-5p expression levels than African Americans (Figure 2B). No significant associations were observed between miR-150-5p levels and the median age at diagnosis (>53, ≤53), tumor size, tumor histology (ductal or lobular), lymph node metastasis, and distant metastasis.

The relationship between miR-150-5p expression levels and survival status (alive/deceased) was assessed in 24 patients with TNBC: 16 (15 alive and one deceased) of these patients showed upregulation of miR-150-5p, and eight (four alive and four deceased) showed miR-150-5p downregulation. Although not significant, reduced overall survival was observed in patients with lower miR-150-5p expression (HR = 0.153, CI 95%, 0.02–1.27, *p* = 0.0610). The same trend was observed in TNBC patients from the TCGA cohort (HR = 0.42, CI 95%, 0.15–1.17, *p* = 0.088), and a significant association was observed in patients with TNBC in the METABRIC cohort (HR = 0.43, CI 95%, 0.27–0.67, *p* = 0.00014; Figure 2C).

### 3.3. MiR-150-5p Is Upregulated in TNBC Cell Lines Compared with Non-TNBC and Non-Tumor Breast Cell Lines

MiR-150-5p endogenous expression levels of the seven breast cancer cell lines evaluated (MCF-7, BT549, HCC1806, MDA-MB-157, MDA-MB-231, MDA-MB-453, and MDA-MB-468) and of the non-tumorigenic breast cells (MCF-10A) were evaluated by RT-qPCR. MiR-150-5p expression levels were significantly higher in four of the six TNBC cell lines than in MCF-10A: BT549 (FC = 3.07), HCC1806 (FC = 1.84), MDA-MB-231 (FC = 4.46), and MDA-MB-453 (FC = 6.72). TNBC cell lines with the lowest expression levels of miR-150-5p were MDA-MB-157 (FC = 0.90) and MDA-MB-468 (FC = 0.75). The non-TNBC cell line evaluated, MCF-7, presented lower levels of miR-150-5p expression when compared to the MCF10A (FC = 0.82) (Appendix A). Considering their expression levels, transfection efficiency and high metastatic potential, the HCC1806 and MDA-MB-231 TNBC cells were selected for the functional analysis.

### 3.4. Inhibition of MiR-150-5p Expression Reduces Cell Proliferation and Clonogenic Growth in TNBC Cells

A significant 34.6% decrease in cell proliferation was observed in HCC1806 cells after 72 h and 96 h of transfection with the miR-150-5p inhibitor (Figure 3A). Consistently, a decrease by 63.75% in the number of colonies was observed with the inhibition of miR-150-5p expression (Figure 3C). Cell proliferation and clonogenicity of the HCC1806 cells were, however, not affected with the ectopic expression of miR-150-5p (Figure 3A,C). In the MDA-MB-231 cells, manipulation of miR-150-5p expression did not affect cell proliferation after 72 h or 96 h (Figure 3B).

### 3.5. Inhibition of MiR-150-5p Expression Reduces Cell Migration and Resistance to Doxorubicin in TNBC Cells

Wound healing assays were conducted to determine the effects of manipulation of miR-150-5p expression on cell migration in HCC1806 and MDA-MB-231 cell lines. MiR-150-5p expression inhibition significantly decreased the percentage of wound closure by 30.68% and 26.12% in HCC1806 and MDA-MB-231 cells, respectively (Figure 4A–D). It is interesting to note that in the MDA-MB-231 cells, the changes in this phenotype were not associated with correspondent changes in their cell proliferation rate. Cell migration was not affected by the ectopic expression of miR-150-5p.

The cytotoxic effects of doxorubicin were evaluated in the miR-150-5p transfected TNBC cells (HCC1806). Cell viability was measured 24 h, 48 h, and 72 h after treatment with 50 nM doxorubicin. At all treatment time points, cell viability was significantly lower in cells transfected with miR-150-5p inhibitor than in NC cells (Figure 4E).

### 3.6. Inhibition of MiR-150-5p Expression Reduces the Expression of MYB and Members of the ERK1/2 Pathway

To determine the effects of miR-150-5p manipulation on the expression of direct mRNA targets (Figure 4F and Figure 5A), we investigated the expression levels of MYB, TP53, ZEB1, and SRC proteins in cells transfected with miR-150-5p inhibitor or mimic. Inhibition of miR-150-5p expression led to increased levels of MYB in the MDA-MB-231 cells (Figure 5B), but not in the HCC1806 cells (Appendix A). No alterations were observed in the expression levels of TP53 and ZEB1 in MDA-MB-231 cells (Figure 5B).

Next, we investigated the expression levels of members of the SRC pathway. In the MDA-MB-231 cells, the manipulation of miR-150-5p expression did not affect the levels of active SRC (p-Src Y416) or inactive SRC (p-Scr Y527). However, the expression levels of p-ERK1/2, which is downstream of the SRC pathway, were reduced upon miR-150-5p inhibition. In addition, increased p-ERK1/2 protein levels were observed with miR-150-5p ectopic expression, suggesting that miR-150-5p plays a role in the activation of the ERK1/2 pathway (Figure 5B). The same changes in expression levels of inactive (p-Scr Y527) and pERK1/2 were also observed for HCC1806 after manipulation of miR-150-5p expression (Appendix A).

## 4. Discussion

TNBC is characterized by aggressive phenotypes, poor outcomes, and limited treatment options. MiRNAs act as post-transcriptional gene expression regulators, suppressing or activating the expression of genes implicated in TNBC development and progression [50,51,52,53]. Mounting evidence suggests that miRNAs can be used to differentiate tumor subtypes [54,55], including breast tumors [56,57,58]. We have previously shown that TNBC and non-TNBC tumors from different patient populations exhibit distinct patterns of miRNA expression [59,60]. Notably, miR-150-5p was one of the miRNAs that were highly expressed in TNBC compared with non-TNBC tumors [59]. In this study, we analyzed breast cancer samples from an independent cohort of 113 patients and confirmed that miR-150-5p expression levels were significantly higher in TNBC tissues than in non-TNBC tissues. ROC curve analysis further supported the ability of miR-150-5p levels to discriminate between TNBC and non-TNBC tumors. In addition, we showed that miR-150-5p expression presented a high tumor specificity, as miR-150-5p expression levels were significantly higher in tumor tissues than in ANT tissues. Previous studies yielded conflicting data regarding the expression of miR-150-5p in TNBC. Our results are in agreement with Lu et al. [28], but contradict reports of lower expression of miR-150-5p in TNBC tissues compared with non-tumor tissues [27,29]. These opposite expression patterns may reflect the ability of miR-150-5p to regulate multiple gene targets, some of which have opposing oncogenic or tumor-suppressive functions [61]. The combined suppression or activation of other miRNAs regulating the same targets or functional pathways can result in a more pronounced effect of a single miRNA with either an oncogenic or tumor-suppressive function in a given type of tumor [62].

Analysis of the relationship between miR-150-5p expression levels and the clinical and histopathological characteristics of patients revealed that miR-150-5p levels were significantly associated with grade 3/4 tumors (TNBC and non-TNBC combined and TNBC subtype only). In contrast to the findings of other studies, no significant association was observed between miR-150-5p expression and other poor prognostic factors, such as lymph node metastasis and distant metastasis [29,63]. Interestingly, survival analysis in the TNBC subgroup showed that miR-150-5p upregulation was associated with prolonged overall survival. These findings were confirmed in patients with TNBC from the TCGA and METABRIC databases and highlight the inconsistent implications of miR-150-5p expression in prognosis, as reported by Wang et al. [14].

MiR-150-5p expression levels were also significantly associated with ethnicity in the entire cohort and in the TNBC and non-TNBC subgroups, with Caucasians showing higher miR-150-5p levels than African Americans. In contrast, our previous global miRNA profiling study [59] showed that African American patients with TNBC presented higher expression levels of miR-150-5p than non-Hispanic White patients. It is worth pointing out that in this study, ethnicity was self-reported and not determined by ancestral genomic profiling; therefore, the ethnicity of the patients may have affected miRNA expression in a way that could not be determined [64,65]. These results support the hypothesis that biological differences exist in miRNA expression patterns between populations and that these differences may influence prognosis and treatment outcomes.

The expression levels of miR-150-5p also varied among cell lines. Most TNBC cell lines had higher miR-150-5p levels than non-tumorigenic breast cells (MCF-10A) and non-TNBC cells (MCF-7). MiR-150-5p expression was manipulated in the highly metastatic HCC1806 and MDA-MB-231 cells [66,67], to evaluate its potential role in modulating TNBC aggressive phenotypes. MiR-150-5p expression inhibition led to a decrease of the TNBC cells tumorigenicity, as significant decreases in cell proliferation, capacity of colonies formation, cell migration, and resistance to doxorubicin were observed. However, ectopic expression of miR-150-5p did not affect these phenotypes. These results suggest that additional levels of miR-150-5p in TNBC cell lines are not required to confer further aggressiveness. Furthermore, a given miRNA, such as miR-150-5p, very likely does not entirely control a given tumorigenic phenotype, requiring the cooperation of other miRNAs that act cooperatively in the same biological process and/or signaling pathway they are involved, by regulating common mRNA targets [62]. Several miRNAs from common clusters or families share similar functions with miR-150-5p and regulate the same mRNA targets. For example, miR-769-5p is also mapped at 19q13 and regulates epithelial-mesenchymal transition (EMT), invasion, and metastasis [68,69,70]. In addition, long non-coding RNAs (lncRNAs) can interact with miRNAs, allowing them to exert their full cellular action. Several studies have demonstrated a relationship between lncRNAs and miR-150-5p in breast tumorigenesis; these lncRNAs include FOXD2 AS1 [31], RNA 01121 [25], and MAFG AS1 [26].

It is also of interest to note that the miR-150-5p impact on tumor aggressiveness was distinct in the patients’ TNBC tissue samples and in the TNBC cell line models. In the patients’ tumor tissues, high miR-150-5p expression levels was associated with higher survival rates, which as mentioned above, was also observed in the TNBC patients from the TCGA and METABRIC databases. At the other hand, in the TNBC cell lines, miR-150-5p behaved most compatible with an oncogenic mode of action, where its inhibition was associated with a decrease in tumor aggressiveness. Several reasons can account for these discrepancies, including the fact that the TNBC clinical samples were derived from patients’ primary tumors and the TNBC cell lines from highly metastatic tumors. However, even within metastatic samples and metastatic cell lines, genotypic and phenotypic discrepancies can occur. In a study conducted by Liu et al. (2019) [71], a comparative analysis of publicly available genomic and transcriptomic signatures of clinical metastatic breast tumors and metastatic breast cell lines models demonstrated substantial differences among these intrinsic signatures, which can affect distinct cancer associated genes’ networks and also affect the classical tumor phenotypes.

Cell adhesion and adherent junctions were the process mostly observed for miR-150-5p functional enrichment analysis. These are critical processes of EMT, which is essential for cell migration and invasion [72,73]. EMT is regulated by several transcription factors and proteins, including E-cadherin, SLUG, Vimentin, SNAIL, ZEB1, and Twist [74]. Alterations in these proteins directly affect the cells tumorigenic behavior by promoting alterations in the cytoskeleton, adhesion, and cell polarity [75]. In this study we evaluated the role of miR-150-5p in the expression of the EMT markers SNAIL, SLUG, Vimentin, and ZEB1. We observed that the expression levels of the early EMT markers SNAIL and SLUG were reduced with the inhibition of miR-150-5p expression. Additionally, significantly higher levels of the oncoprotein MYB (a direct miR-150-5p target) was observed with the inhibition of miR-150-5p expression. Ectopic expression of miR-150-5p, however, did not change MYB expression levels, which were similar to those observed in the NC cells.

Previous studies on lung, gastric, cervical, and breast cancers have shown that miR-150-5p promotes cell proliferation and migration by targeting the SRC kinase signaling inhibitor 1 (SRCIN1), a negative regulator of *SRC* and an experimentally validated direct target of miR-150-5p [23,28,76]. SRCIN1 inhibits SRC via phosphorylation (tyrosine 597) and the subsequent activation of CSK. Interestingly, a relationship between pERK and SNAIL has been previously reported in breast cancer, with high levels of pERK and SNAIL being associated with increased migration [77]. Therefore, we hypothesized that inhibition of miR-150-5p, which reduced the expression levels of SNAIL in TNBC cells, would negatively affect the transcriptional activity of SRCIN1 and other proteins of the SRC pathway, such as pERK and pFak. These proteins are regulated by the phosphorylation of SRC and promote cell migration and invasion. Lower expression levels of pERK1/2 were observed with the inhibition of miR-150-5p; conversely, ectopic expression of miR-150-5p led to an increase in pERK1/2 levels. In contrast, the levels of active (p-Src Y416) or inactive (p-Scr Y527) SRC or pFak were not affected with the inhibition and ectopic expression of miR-150-5p. Manipulation of the expression levels of these proteins would be necessary to show the reverse activity of miR-150-5p function in these cells and the reversal of the tumorigenic phenotypes.

## 5. Conclusions

Our findings suggest that miR-150-5p is involved in TNBC aggressiveness, presenting differential expression in these tumors compared with non-tumor and non-TNBC tissues samples. Our results also indicate that miR-150-5p regulates cell proliferation, clonogenic growth, drug resistance, and migration as observed in the two TNBC cell lines evaluated, HCC1806 and MDA-MB-231. Although several mechanisms can underlie the tumorigenic functions of miR-150-5p in these cells, changes in the expression of its target genes *MYB*, *ZEB1*, and members of the *SRC* pathway can contribute to its oncogenic role in TNBC.

## Figures and Tables

**Figure 1 cancers-14-02156-f001:**
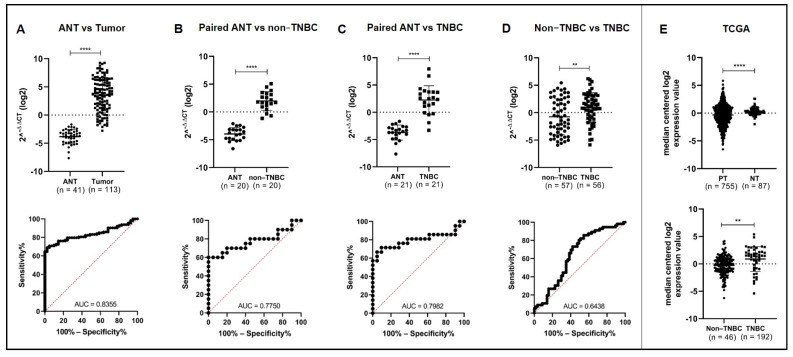
Expression levels (upper panel) by RT-qPCR and ROC/AUC values (lower panel) of miR-150-5p in the group of tumors and ANT tissues evaluated. Higher expression levels of miR-150-5p were significantly observed in the tumor tissues vs. ANT (**A**), paired ANT vs. non-TNBC (**B**), paired ANT vs. TNBC (**C**), and non-TNBC vs. TNBC (**D**). Significant up-regulation of miR-150-5p expression levels by RNAseq in from the TCGA breast cancer cases were observed in primary breast tumors when compared to normal breast tissue ((**E**) upper panel) and TNBC vs. non-TNBC cases ((**E**), lower panel). ANT: adjacent non-tumor tissue; PT = primary tumor; NT = normal tissue. Statistical significance ** *p* < 0.01, **** *p* < 0.0001.

**Figure 2 cancers-14-02156-f002:**
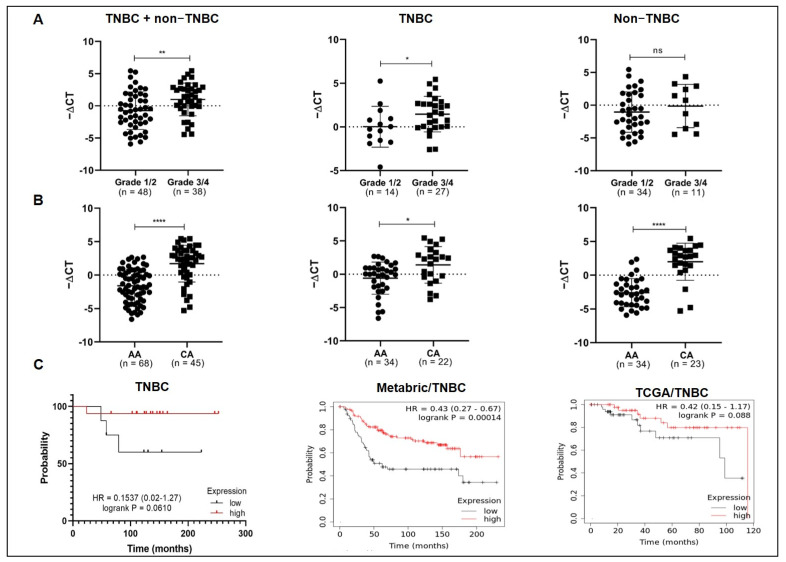
Association of miR-150-5p expression levels with tumor grade, ethnicity, and overall survival in TNBC. Relationship between miR-150-5p expression levels and tumor grade (**A**) and ethnicity (**B**) in the entire cohort and in the TNBC and non-TNBC subgroups; significance was determined using the Mann–Whitney *U* test. (**C**) Survival analysis of patients with TNBC (**left**) and of the METABRIC (**middle**) and TCGA (**right**) cohorts. * *p* < 0.05; ** *p* < 0.01, **** *p* < 0.0001; ns: not significant.

**Figure 3 cancers-14-02156-f003:**
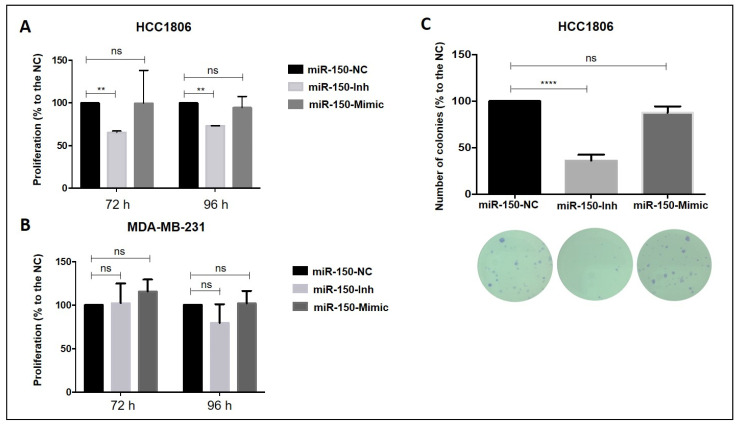
Cell proliferation and clonogenic growth assays in the TNBC transfected cells. (**A**,**C**) Inhibition of miR-150-5p reduced cell proliferation and clonogenicity in HCC1806 cells compared with NC cells. (**A**,**B**) In both cell lines, ectopic expression of miR-150-5p did not affect cell proliferation. Inh: Inhibitor; NC: negative control. ** *p* < 0.01; **** *p* < 0.0001; ns: not significant.

**Figure 4 cancers-14-02156-f004:**
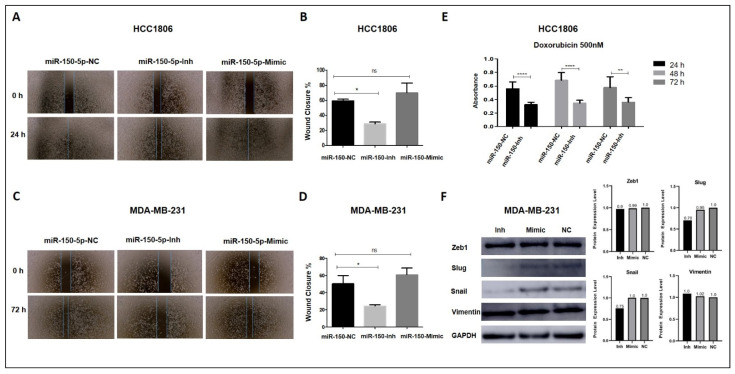
Cell migration and cytotoxicity assays in the TNBC transfected cells. (**A**–**D**) Inhibition of miR-150-5p reduced migration in HCC1806 (**A**,**B**) and MDA-MB-231 (**C**,**D**) cells. In contrast, ectopic expression of miR-150-5p had no effects on cell migration. (**E**) Inhibition of miR-150-5p reduced cytotoxicity of doxorubicin in HCC1806. (**F**) Western blotting analysis and quantitative readings of epithelial-to-mesenchymal transition markers in MDA-MB-231 cells. * *p* < 0.05; ** *p* < 0.01; **** *p* < 0.0001; ns = not significant. NC: negative control.

**Figure 5 cancers-14-02156-f005:**
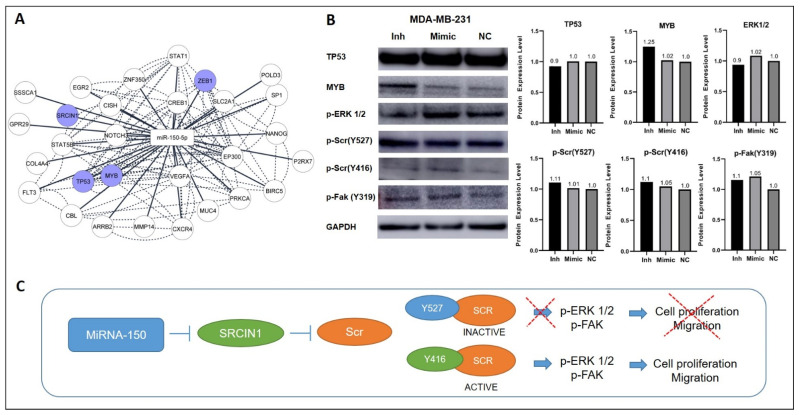
Protein expression analysis of mir-150-5p direct targets and members of the ERK1/2 pathway in the TNBC transfected cells. (**A**) Network of miR-150-5p and experimentally validated target genes (Cytoscape 3.9.1). (**B**) Western blotting and quantitative readings of miR-150-5p mRNA targets and members of the SRC pathway markers in transfected and NC MDA-MB-231 cells. (**C**) Proposed model of miR-150-5p regulation of the SRC pathway. NC: negative control.

**Table 1 cancers-14-02156-t001:** Clinical and histopathological information of the breast cancer patients studied, distributed by the subtype.

	TNBC	Non-TNBC	*p*-Value
Age			
≤53	28	34	*p* > 0.9999
53	19	23	
Tumor Size			
≤2 cm	33	27	*p* = 0.6758
>2 cm	18	19	
Histology			
Ductal	56	5	*p* = 0.0129
Lobular	0	7	
Grade			
1/2	0	29	*p* = 0.0135
2/3	46	19	
Lymph node			
Positive	17	22	*p* = 0.5152
Negative	24	22	
Metastasis			
Positive	3	8	*p* = 0.2177
Negative	41	44	
Race			
AA	34	34	*p* > 0.9999
CA	22	23	

## Data Availability

Publicly available datasets were analyzed in this study. This data can be found here: https://app.box.com/s/h5bc82qa1t25xe3z6cstx876mexy4u2j, accessed on 10 March 2022.

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
