# Peer review of "MiR-150-5p Overexpression in Triple-Negative Breast Cancer Contributes to the In Vitro Aggressiveness of This Breast Cancer Subtype"

_cancers, 2022, doi:10.3390/cancers14092156_

Round 1

Reviewer 1 Report

I think it's a good job. I consider that it is a study with a lot of work, wide and detailed. However, the conclusions should be more specific and perhaps include some material that is currently provided as complementary. Here are some suggestions for improvement.

In section 3.3 they talk about non-TNBC cell lines, I understand that it is MCF-7 but they do not detail or explain the expression levels of miR-150-5p nor in non-tumor cell lines. I understand that they appear in the supplementary material, but it would be appropriate to detail it in the writing. On the other hand, in section 3.4, although they explain the results obtained for both selected TNBC lines, in relation to MDA-MB-231 with the miR-150-5p inhibitor, they do not observe any change in proliferation at any of the determined times, however subsequently observed a decrease in cell migration of the MDA-MB-231 line in the wound closure assay. It would be interesting to explain this point. In the HCC1806 line, all the exposed results are in agreement, reduction of proliferation, migration and formation of colonies. On the other hand, in Figure 4 I would also include the graph of inhibition of miR-150-5p reduced cytotoxicity of doxorubicin and Western blotting analysis and quantitative readings of epithelial-to-mesenchymal transition markers for both cell lines, since the study mainly focuses on them. Finally, I would consider it more appropriate to be more specific in the conclusion. They speak of results on TNBC lines, however the results in relation to proliferation, migration and aggressiveness focus on two of the six initial lines, which I would name in the conclusion. 

Reviewer 2 Report

Sugita et al investigated the expression and functions of the miRNA miR-150-5p in triple-negative breast cancer (TNBC). The authors found that miR-150-5p is expressed at higher level in TNBC compared to non-tumour tissues and non-TNBC. High expression of miR-150-5p is also associated with high tumour grades, the Caucasian race, and a prolonged overall survival. The authors then moved to in vitro experiments with TNBC cell lines to determine the functions of miR-150-5p. The authors found that modulating miR-150-5p level affects cellular proliferation, clonogenecity, cell migration, resistance to doxorubicin, and expression of known direct and indirect targets of miR-150-5p.

While the study is interesting, I have several comments:

Comments:

  1. Line 237: Figure E -> Figure 1E
  2. Lines 283-284: Can the authors explain why they choose MDA-MB-231 and HCC1806 TNBC cells for further analysis?
  3. Figure 4E: Doxorrubicin (in the figure title) -> Doxorubicin.
  4. Figure 4F is not cited in the Results section. The authors will need to describe the experiment and the results.
  5. Figure 5 and Supplementary Figure 2: The authors need to perform western blots of total proteins for ERK 1/2, Scr, and Fak. Any changes in phosphorylation level could be explained by a change in the total protein level and not in a specific change in protein phosphorylation.
  6. Lines 326-328: While I agree that inhibition of miR-150-5p leads to an increase in MYB in the MDA-MB-231 cell line, it is clearly not the case for the HCC1806 (Supplementary Figure 2). The MYB protein level goes from 1 in the NC to 1.04 following inhibition, which can hardly been called an increase, and the experiment has been performed only once.
  7. As treatment with the miR-150-5p mimic does not seem to have any effect in the different experiments, can the authors rule out that the mimic is just not working? Does the authors have any experimental evidence of the mimic working?
  8. Also, it is not clear how the patients’ data and the in vitro data are making sense together. From the patients’ data, it seems that a high level of miR-150-5p is beneficial while the in vitro data are showing that a high level of miR-150-5p is associated with cellular proliferation and migration of cancer cells, not exactly what I would call beneficial. Can the authors comment on this in the Discussion?

Reviewer 3 Report

Dear Authors

The manuscript is very interesting and well written, addressing a hot topic in cancer the microRNAs.

Regarding the manuscript, improvement would like to suggest that race is replaced by ethnicity, in all manuscript. Versus (vs) is Latin therefore must be in italic, correct in all manuscript as well as genes names that must be also in italic. when using et al. the et al. must in italic once it is Latin.

The introduction introduces very well the topic, finishing with the aim of the research. But then the authors wrote the conclusions, which is not the correct place for that, therefore suggest to have only the aim and nothing more.

Material and Methods are very descriptive although have already some results included, therefore suggest removing table 1 from this section and moving to Results.

All abbreviations in the manuscript must before be fully written and only after that use the abbreviation. the abbreviations that need to be corrected are:

RPMI, DMEM, PBS, RIPA, PDV among others, review all manuscript.

All experiences must have a reference to support the technique or method used. The clonogenic assays were really performed like that? without replacing medium at 7 days?

All reagents, equipment and software must have between brackets (producer, City, Country). All antibodies used must have the respective clone.

Highly important: the authors must clearly clarify why in section 2.2 presents one ER+ breast cancer cell line, six TNBC and one non-tumorigenic human breast epithelial cell line, but from a moment only the cell lines MDA-MB-231 and HCC 1806 cells were used. 

Results when using age or months use also the range minimum and maximum and done in line 107.

Discussion well done.

Conclusions are good.

Round 2

Reviewer 2 Report

The authors have answered all my comments.